# Development of an Artificial Vision for a Parallel Manipulator Using Machine-to-Machine Technologies

**DOI:** 10.3390/s24123792

**Published:** 2024-06-11

**Authors:** Arailym Nussibaliyeva, Gani Sergazin, Gulzhamal Tursunbayeva, Arman Uzbekbayev, Nursultan Zhetenbayev, Yerkebulan Nurgizat, Balzhan Bakhtiyar, Sandugash Orazaliyeva, Saltanat Yussupova

**Affiliations:** 1Department of Electronics and Robotics, Almaty University of Power Engineering and Telecommunications named Gumarbek Daukeyev, Almaty 050013, Kazakhstan; gb.univers@yahoo.com (A.N.); n.zhetenbaev@aues.kz (N.Z.); y.nurgizat@aues.kz (Y.N.); orazalieva-sandugash@mail.ru (S.O.); s.yusupova@aues.kz (S.Y.); 2Academy of Logistics and Transport, Almaty 050012, Kazakhstan; g.balbayev@gmail.com; 3Department of Information Security, Eurasian National University, Astana 10000, Kazakhstan; 4Research Institut of Applied Science and Technologies, Almaty 050013, Kazakhstan; a.uzbekbayev@su.edu.kz; 5Department of Heat Power Engineering, NCJSC S.Seifullin Kazakh Agro Technical Research University, Astana 10000, Kazakhstan; bahtyarbalzhan@gmail.com

**Keywords:** machine-to-machine communication, delta robot manipulator, artificial vision system, RGB, MASK-R-CNN

## Abstract

This research focuses on developing an artificial vision system for a flexible delta robot manipulator and integrating it with machine-to-machine (M2M) communication to optimize real-time device interaction. This integration aims to increase the speed of the robotic system and improve its overall performance. The proposed combination of an artificial vision system with M2M communication can detect and recognize targets with high accuracy in real time within the limited space considered for positioning, further localization, and carrying out manufacturing processes such as assembly or sorting of parts. In this study, RGB images are used as input data for the MASK-R-CNN algorithm, and the results are processed according to the features of the delta robot arm prototype. The data obtained from MASK-R-CNN are adapted for use in the delta robot control system, considering its unique characteristics and positioning requirements. M2M technology enables the robot arm to react quickly to changes, such as moving objects or changes in their position, which is crucial for sorting and packing tasks. The system was tested under near real-world conditions to evaluate its performance and reliability.

## 1. Introduction

Presently, the widespread adoption of artificial intelligence (AI) with various robotic systems to mimic human intelligence has triggered tremendous changes in today’s technological environment and human activities as a whole. Over the past decades, elements of artificial intelligence, using techniques such as machine learning, deep learning, cognitive computing, etc., have made significant progress in creating a variety of applications and robotic systems that can perceive, analyze, and process incoming information, including making decisions autonomously without human intervention, in various fields such as logistics, transport, surveillance systems, automated complexes, health care, and science [1]. 

The concept of applying artificial intelligence is justified as an interdisciplinary and complex technology to create intelligent robotic systems. This is justified by the fact that it combines different technologies that can operate as independent systems and also in combination with various Internet of Things (IoT) devices, allowing remote control of systems or devices through mobile control platforms. The complexity of intelligent robotic systems technology also lies in the integration and interaction between control, monitoring, and automation systems, including centralized data storage and processing and production facilities, which in turn must have properties such as self-awareness, self-prediction, self-comparison, self-configuration, self-service, organization of the process being performed, and a sufficient level of sustainability without human involvement in the intelligent production process. This trend in the formation of intelligent robotic systems leads to the development of new approaches to industrial production based on the process of generation, as well as processes using AI elements of machine vision and computer vision, which raises the relevance of research in the field of generation and detection of objects using elements of artificial vision without human participation. 

In a broad understanding, the main role of artificial vision in modern intelligent robotic systems is to interpret obtained visual data, including processes such as object recognition, tracking, and image segmentation [2]. Currently, the effectiveness of image classification, object detection, and segmentation algorithms strongly depends on the analysis of visual data and its spatial patterns, including real-time data processing. The integration of complex data acquisition systems, processing algorithms, and computational complexity requires significant time and computational resources [3].

Various image augmentation techniques can be applied to simplify the process and prevent over-learning. These methods include modifications such as rotating, cropping, rotating, resizing, adding noise, random erasure, combining images, and other similar techniques. However, most robotic systems have insufficient data storage capacity and computational power, which makes the solution to this problem more urgent. 

The combination of artificial intelligence algorithms with Internet of Things (IoT) devices is also popular. This approach is based on deployed deep or machine learning models that allow for the processing and analysis of incoming visual data, including making decisions collaboratively or independently without human intervention [4]. In this case, storage in the form of a cloud server equipped with neural networks is introduced into the Internet of Things environment to perform the functions of load balancing of visual data for storage and processing, as well as to perform computational operations, which contributes as an assistant in analyzing the input data received from intelligent devices. This kind of interconnection between IoT devices and various servers is defined as “machine-to-machine communication” (M2M), which today is becoming more dominant than the interconnection between a human and an intelligent device or machine.

In today’s world, artificial vision and machine-to-machine communication play vital roles in the development of robotics and the automation of industrial and domestic processes. Artificial vision, as a key component of artificial intelligence, gives robots and automated systems the ability to perceive and analyze their surroundings using cameras, sensors, and other devices. M2M, on the other hand, allows these systems to share data and information, opening up new possibilities for network coordination and collaboration. Despite significant advances in artificial vision and machine-to-machine communication, many challenges in the development of intelligent robotic systems still exist for developers and researchers.

The relevance of this study is due to several important factors that take into account the progress of robotic systems development. First, the current trend towards the automation of manufacturing processes and the increasing requirements for accuracy and efficiency stimulate the development of more advanced control systems for robotic systems. Second, the application of M2M technologies greatly simplifies the integration of various components of a robotic system, increasing its flexibility and scalability in space. Third, advances in artificial vision and machine learning open new perspectives for creating adaptive control systems that can effectively respond to changing production conditions and automatically adjust the operation of the robotic system. In this regard, the development of an artificial vision system for a parallel manipulator using M2M technologies is an urgent and promising task that can bring significant changes in the control processes of robotic systems used in the production process.

This research aims to develop an artificial vision system for a robotic device like the delta robot—a manipulator using machine-to-machine technologies. Specific objectives of the research include studying the possibilities and advantages of M2M technologies for delta robot manipulator operation, developing algorithms for image processing and data analysis, and creating an intelligent control system to coordinate the actions of the delta robot manipulator based on the information received from the artificial vision system. The research results can be applied in various fields, including manufacturing, medicine, service, and education.

In the course of the research implementation, the authors review the latest trends and achievements in the development of artificial vision systems in robotic systems using M2M technologies. Also, this paper details the principles of the combined operation of the artificial vision system and machine-to-machine communication, including their role in the creation of autonomous robots and intelligent control systems based on the developed prototype delta robot manipulator by the authors [5].

In [5], the authors created a robotic system consisting of a fixed and mobile platform with three axes of freedom to move in a limited space, perform grasping operations, and function at high speeds. Delta robot arms of this type have found wide application in the industrial sector, in areas such as the sorting of objects/parts, assembly processes, and other operations where fast movement in three-dimensional space is a key factor. Today, they are also used in medical and pharmaceutical processes, as well as in the production of electronics and other high-precision products.

Considering the peculiarities of the positioning of the proposed flexible robot and the key forward and inverse kinematic parameters, the development proposed in the paper by the authors of [5] was intended to conduct research on the creation of a machine vision system with an intelligent control system for its further optimal positioning and application in sorting processes.

This research on the development of an artificial vision system for a delta manipulator robot includes several key steps necessary to achieve the objective:Development of computer vision algorithms for object recognition and image processing. This stage includes the development of algorithms for processing images obtained from the camera located in the delta robot manipulator, realized by studying algorithms for object detection, pattern detection, and motion identification. For this purpose, this paper applies the methods of machine learning and computer vision.Development of a delta robot manipulator control system. In this stage, a control system is developed that will use image processing algorithms to control the motion and positioning of the robot. The control system developed should be able to take input data from the camera, process it using algorithms, and then send commands to move the robot.Integration of the artificial vision system with the M2M-based robot control system to fulfill the specified tasks. This stage involves integrating the created artificial vision algorithm with a complex delta robot arm control system. The integration of the systems should be able to control the robot’s motion in such a way that it can respond to visual stimuli with a high speed and accuracy while respecting the forward and backward kinematics of the robot’s operation.Testing and debugging the artificial vision system in real robot working conditions. This stage includes the experimental study of the delta robot manipulator.

The significance of the structure proposed by the authors, combining the artificial vision system for the parallel manipulator with the use of M2M technologies, lies in the significant improvement of the performance and accuracy of the parallel manipulator. The use of an artificial vision system increases the autonomy and efficiency of the manipulator, enhancing its ability to adapt to different conditions and tasks, considering data processing using M2M technology.

This research has great practical significance in the field of creating affordable mobile robotic systems to simplify the production process in sorting parts, objects, garbage, etc. The results obtained from the research can find application in various industries, including manufacturing process automation.

It is also important to note that the focus of this work is on the development of an affordable robotic system with an emphasis on the integration of low-cost technologies that will increase productivity and quality of the manufacturing process, reducing the time required to complete sorting tasks and minimizing the likelihood of errors. In addition, such a system has the ability to operate around the clock without interruption, which will reduce personnel costs and increase overall production efficiency. As a result, the solution proposed by the authors can help industries cope with labor shortages and improve their competitiveness in the market.

## 2. Related Work

Automation of production processes using robotic systems provides many advantages in replacing the role of humans in reliable detection and rapid manipulation of various production processes [6]. To date, the research interest of scientists around the world is focused on developing intelligent robotic systems that can be used in various types of industrial and domestic activities, the effectiveness of which is confirmed by the lower consumption of resources for their maintenance and training processes. The most important factor in the development of intelligent robots lies in ensuring high reliability and flexibility of the system being developed, as well as in creating an effective system of adaptation to different circumstances and requirements. In this case, the introduction of artificial intelligence into the robotic system helps developers increase productivity by reducing energy consumption for data processing and further decreasing the time required to perform various operations in real-time [7,8].

This trend in the development of intelligent robotic systems is implemented using machine learning algorithms and deep learning artificial neural networks to thoroughly understand human behavior to solve complex tasks by robots in industrial and domestic processes. Also, the integration of robotic systems with artificial intelligence today allows robots to perform such complex tasks as the detection of objects, their recognition, and segmentation by various attributes, including the processing of a large amount of data with further intellectual analysis to form a consistent action. In this case, the above standard tasks related to object detection and segmentation, including the processes of their tracking, are performed using a combination of artificial vision and machine learning [9,10].

The above integration of robotic systems with various computer vision capabilities allows scientists to create fully functional intelligent robots using a variety of algorithms and performance parameters of artificial intelligence and machine-to-machine communication. This is due to the high increase in demand for the Internet of Things with the presence of the application of artificial vision capabilities over the last decade in the industrial sector to optimize image-based inspection processes, object digitization, and object detection, as well as to perform visual maintenance and robot calibration, including mobile navigation of the robotic system in space [11]. It is also important to note that artificial vision methods and algorithms play a key role in the control systems of robotic systems, as they allow robots to interact with their environment, analyze it, and make decisions based on the information obtained.

The artificial vision hierarchy in the implementation includes three main steps that aim to endow the robotic system with the ability to interpret and analyze visual information, similar to human perception. These steps are as follows:

Image Classification: this process involves the ability of computer vision to learn and recognize graphic images, ultimately assigning a category according to pre-configured preset sectors or object labels, focused on certain types of objects and criteria. This process is fundamental in the development of an artificial vision system, as the system must have an extensive range of images that vary according to different parameter selection criteria, such as object shape, color, size, etc.

Object Detection: this process involves the automatic identification and finding of the spatial location of objects using computer vision methods and machine learning algorithms, bounding box methods, as well as the identification of key points in the image. This forms a spatial understanding of the localization of objects in the image, where a high level of detection accuracy is required.

Segmentation: this is the process of dividing an image into several parts or segments, where each pixel is assigned to a particular object or background category, marking the boundaries of each object. It is based on the processes of selecting and identifying objects or regions in an image, realized by applying pixel-based segmentation and semantic segmentation processes. The segmentation process is an important tool in an artificial vision system, as it enables the correct interpretation of the environment, increasing the accuracy and efficiency of the recognition process. It also improves the visual understanding of the image content by highlighting the relationship with the different elements of the background under consideration.

Considering the hierarchy of the artificial vision system structure, the visual overview of an artificial vision-based robotic system combined with machine learning techniques facilitates the kinematics of the robot, which significantly impacts its basic performance and physical abilities.

It is important to note that target detection in a robotic system is implemented using one-stage or two-stage artificial vision detection to identify and locate objects in images or video files using various algorithms to analyze visual patterns and select objects against the background of space [11].

Artificial vision includes many algorithms and analysis techniques depending on the application domain and performance, such as convolutional neural networks (CNN), the support vector method (SVM), feature-based object detectors (Haar, HOG, SURF, etc.), segmentation methods (such as U-Net, Mask R-CNN), competitive image and keypoint detection methods (SIFT, ORB, SURF, etc.), deep neural networks for image generation (GAN, VAE), etc.

Also, the most commonly used CNN architectures are MobileNetV2, ResNet50, and DenseNet121, which were applied by the authors [12] in a previous study. In this study, the authors achieved a 2–3% increase in accuracy rates, achieving 88% for MobileNetV2, 91% for ResNet50, and 92% for DenseNet121 by adding a squeeze-and-excitation (SE) block that allows the neural network to better extract important features. This study, along with the study by the authors [13], demonstrates that the use of modern machine learning and deep learning techniques combined with advanced neural network architectures can not only improve the accuracy of detecting abnormal situations depending on various factors but also significantly accelerate the data processing process.

Another study [14] achieved a 96% detection and classification accuracy using detection transformer computer vision (DETR) to manage defects in the supply chain. The authors also improved the system’s performance through artificial intelligence algorithms to detect defects in real-time. However, such a system requires a significant amount of data and computational power and also requires the consideration of many factors such as the location of objects, lighting conditions, distance to objects, and other factors.

In their work, the authors of [15] dedicated algorithms to image processing such as Harris, SUSAN, FAST, SIFT, and SURF, emphasizing the importance of illumination conditions in the considered space, which can ultimately reduce detection accuracy in case of large changes in illumination levels. Despite this, a stable algorithm with a matching accuracy that can reach over 94% has been experimentally achieved.

A new efficient feature extraction algorithm for multispectral images based on geometric algebra, GA-ORB, is presented in [16]. Experimental results confirmed the effectiveness of the GA-ORB algorithm, which outperforms a number of previous methods in terms of distinguishability and reliability in extracting and matching points of interest. In addition, this algorithm is significantly faster in computation. Also, in the research review article “A Review of Machine Learning and Deep Learning for Object Detection, Semantic Segmentation, and Human Action Recognition in Machine and Robotic Vision” the authors [17] emphasize key aspects of deep learning architectures in robotic vision, including convolutional neural networks (CNN), recurrent neural networks (RNN), and generative adversarial networks (GAN) in their framework architectures.

The above technologies are widely used for tasks such as object detection, pose estimation, and semantic segmentation. CNNs are central to the tasks of object detection, image classification, and scene segmentation. They efficiently extract complex features from raw image data, enabling accurate object identification and tracking. In tasks requiring temporal sequence analysis, RNN, particularly long short-term memory (LSTM) networks, are especially important. These networks are excellent at tracking moving objects and predicting future actions based on historical data. It is also important to note that artificial vision methods and algorithms play a key role in the control systems of robotic systems, as they allow robots to interact with their environment, analyze it, and make decisions based on the information obtained.

No less important is the formation of the control system of the robotic system, which requires a comprehensive approach that includes requirements analysis, design of the control structure, selection of technologies and tools for structure development and data processing, as well as testing and optimization of the created structure to achieve high performance and reliability. In this case, the consideration of these criteria makes the delta robot manipulator control system a powerful tool for automation and optimization of manufacturing processes, providing high performance, safety, and resource savings.

The authors of [18] investigated the process of improving the motion control system of a parallel delta robot depending on complex conditions such as high nonlinearity and uncertainty instability. The proposed methods and approaches to create robust control of a nonlinear delta-parallel robot allowed for more accurate and robust control of the robot’s motion, improving its performance and reliability in various operating environments.

Similarly, the authors of [19] applied a programmable gate array (FPGA) to develop a control system for the delta robot with three pneumatic actuator subsystems, achieving high accuracy in the available workspace for controlling the three-dimensional trajectory tracking of the delta robot.

In “Modeling, Design, and Control of a 4-Arm Delta Parallel Manipulator Employing Type-1 and Interval Type-2 Fuzzy Logic-Based Techniques for Precision Applications,” the authors of [20] created a control strategy based on Fuzzy PD and Fuzzy PID configurations.

Based on the above sources, as well as studies [21,22,23], the control system of a delta robot manipulator should allow for customization of the robot’s movements depending on the task and working conditions. This includes motion control within a specific region of the workspace, providing flexibility and adaptability of the robot to different factors and conditions. It is also important when creating a control system to consider integration with other automated systems and components, which allows for the creation of complete solutions for automating various processes. This increases efficiency and productivity while minimizing energy consumption. Artificial vision methods and algorithms play a key role in the control system of a robotic system because they allow the robot to interact with the environment, analyze it, and make decisions based on the obtained information. The interrelation of existing methods and algorithms of artificial vision with the control system of the robotic system allows for processing and analysis of the data received by sensors, identifying patterns, contours, and characteristics of objects, which is the basis for decision-making by the robot.

Despite the diversity of a wide sector of existing artificial vision methods and algorithms, most of the authors give preference [24,25,26,27,28] to the CNN model, which has established itself as one of the most powerful and widely used methods in the field of artificial vision. CNN algorithm models allow a robotic system to automatically extract hierarchical features from images or videos, which makes them effective for classification tasks, object detection, segmentation, and many others. Here, the authors of [29] use software libraries such as Tensor Flow 2, ImageAI 2.3.0., GluonCV 0.9.0, and YOLOv7 in implementing the algorithms for target detection.

Also, in developing “A Vision-Based Path Planning and Object Tracking Framework for a 6-DOF Robotic Manipulator”, the authors of [30] applied a triangulation technique in a three-dimensional stereovision coordinate system (SVCS) with an embedded RGB marker system, which eventually increased the accuracy of robot path prediction up to 91.8% by combining learning models with a color region-tracking process and machine learning.

In creating joint identification and tracking modules in their work, researchers [31] applied the hyper frame vision (HFV) architecture consisting of a 3D sensor. However, the continuous process of capturing a sequence of stereo images using the camera is buffered into a large frame memory, which is energy consuming for a robotic system, which in turn requires processing the acquired large data in real-time.

To ensure the efficiency of the results, the authors [32,33] in their work applied the R-CNN, FASTER-RCNN, and MASK-RCNN algorithms, among which the last one has the highest target detection accuracy of up to 89.9%, which is estimated to be the most efficient among the considered algorithms.

Neural networks are superior to other methods and solutions in the task of object detection due to their ability to provide more accurate results in the shortest possible time for a robotic system. An intelligent robotic system developed based on the application of a neural network with artificial intelligence is a modern technological solution that allows for training the system to adapt to a variety of situations, make decisions, and perform tasks with minimal human participation, as well as to increase their performance significantly and the level of autonomy and accuracy of tasks. This is confirmed by the practical achievements of the authors mentioned above.

The robotic system studied in this paper belongs to a type of delta robot manipulator that has wide applications in manufacturing processes to perform various operations such as parts assembly, packaging, and precise positioning of objects. When building complex integrated robotic systems such as the delta robot arm, it is necessary to consider basic requirements such as the following [34]:High accuracy and speed of movement: delta robots must a have high accuracy and speed of movement, so the intelligent control system must ensure fast and accurate execution of given commands.Stable, wear-resistant design: the delta robot arm must be rigid to ensure stable and accurate operation. The design should be strong and resistant to deformation.High reliability: the control system must be robust and stable to minimize the likelihood of failure and ensure continuous operation of the manipulator.Programmability and flexibility: the control system should support programming and adjustment of the manipulator parameters for different tasks and working conditions.Safety: When designing and operating delta robots, safety measures must be taken into account to prevent possible injuries and damage.Ease of use: the control system should be intuitive and easy to use to facilitate the operator’s work and increase the manipulator’s efficiency.Ability to integrate with other systems: the control system should be compatible and integrate with other automated systems to perform complex tasks and increase productivity.

Based on the literature review of research in this area and the requirements, the authors propose a methodology for applying a convolutional neural network (CNN) based the artificial vision algorithm for delta robot manipulators based on an additive RGB visual spectrum model. In this study, the components of the computer vision system and the control system of the delta robot manipulator are realized using machine-to-machine communication using data mining techniques between electronic boards, providing effective control behind the computer vision system and the manipulator.

The computer vision system in this case is used to recognize objects and determine their position and orientation in space. These data are transmitted to the electronic boards of the delta robot control system, which perform trajectory calculations and commands to perform the task of manipulating the object.

M2M-based data mining techniques allow the system control process to be optimized, taking into account various factors such as speed and accuracy of task execution, power consumption, etc. This in turn makes it possible to improve the efficiency of the overall system and ensure accurate and fast task completion. 

In such a context, the importance of machine-to-machine communication is to ensure that the data mining techniques of the delta robot control system work together with the components of the computer vision system, which will shape the efficient execution of the assigned tasks.

Addressing the above challenges with compliance in building complex integrated robotic systems, the research proposed in this paper is distinctive and fills the gaps in the following aspects:This research focuses on the integration of an artificial vision system with machine-to-machine (M2M) communication technology, which improves real-time device interaction, enabling faster and more accurate manipulation of a parallel robot in response to changes in the environment.Unlike the use of simple algorithms for object recognition, the use of MASK-R-CNN allows for high accuracy and detection in identification and positioning processes.An important contribution is the adaptation of the data obtained from MASK-R-CNN to the delta robot control system. This takes into account the creation of unique characteristics and positioning requirements of the delta robot.System testing was performed under conditions that are as close to real-world conditions as possible. This allows for a more accurate assessment of the effectiveness and reliability of the developed system.The combination of an artificial vision system with M2M technology in this study allows for a significant increase in the speed of operation and overall performance of the robotic system. This metric is critical for tasks such as sorting and packing parts, especially in dynamic or constrained environments.

The proposed solution in this study not only integrates state-of-the-art technology to create a more efficient robotic system but also provides practically applicable solutions that can significantly improve the current methods of automating manufacturing processes.

## 3. Materials and Methods

The design and research of a fully functional delta robot manipulator in this paper was carried out in four stages:(1)Solving the kinematic problems of manipulator construction solved earlier by the authors [5] in the study, “Trajectory Planning, Kinematics, and Experimental Validation of a 3D-Printed Delta Robot Manipulator”.(2)Development of the mechanical design and electronic system of the delta robot manipulator.(3)Creation of an artificial vision system for the prototype delta robot manipulator and its machine-to-machine communication protocol.(4)Experimental study of the created artificial vision system with M2M interactions in the real working conditions of the delta robot manipulator.

### 3.1. Solution of Kinematic Problems of Delta Robot Manipulator Construction

In this study, a type of parallel robot, the delta robot, was chosen as the prototype manipulator. The typical design of this type of delta robot is shown in Figure 1. It consists of the following elements: two platforms, a fixed upper base (1), and a small mobile platform (8) connected by three arms. Each arm comprises two parts: the upper arm (4) rigidly connected to the motor (3) located on the upper base, and the lower arm, which forms a parallelogram (5a, 5b) with so-called universal joints (6, 7) at its corners, allowing for angle adjustments. Each parallelogram is connected to the upper arm by a joint (16) in such a way that its upper side always remains perpendicular to its arm and parallel to the plane of the upper base. This configuration ensures that the movable robot platform, attached to the lower sides of the parallelograms, will also always be parallel to the upper base. The position of the platform can be controlled by changing the angle of rotation of the upper arms relative to the robot base using motors.

In the center of the lower platform (8), the so-called robot work unit (9) is attached. This is a manipulator with a gripping device. If necessary, another motor (11) can be additionally used, which provides rotation of the working body via a boom (14). Based on a typical model of a delta robot (Figure 1), a 3D model of the delta robot manipulator shown in Figure 2 was created using SolidWorks for further printing on a 3D printer, the Flying Bear Ghost 4S. 

The delta robot consists of two triangular platforms, one stationary (1), and the other movable and an end effector (4). The platforms are connected by three kinematic chains, each consisting of two active (2) and passive arms (3) [5].

The main advantage of the developed delta robot manipulator is the high speed of the fulfillment of set operations due to the placement of weighty motors on the base of stationary type. In this case, the moving mechanisms are arms and the bottom platform, which were made of light composite materials to reduce their force of inertia. 

To build the prototype, the kinematic problem was solved when the desired position for the robot arm was known. This involved determining the angle values required to rotate the motors connected to the delta arms of the robot arm, followed by adjusting the gripping processes and confirming their correct positioning. This process of determining the angles for the kinematic problem is known as the inverse kinematic problem. The fixed base of the robot and its moving platform can be represented as equilateral triangles in Figure 3, where the angles of rotation of the robot arms concerning the base plane (angles of rotation of the motors) are denoted as *Ѳ*_1_, *Ѳ*_2_, and *Ѳ*_3_, and the coordinates of the point E_0_ located in the center of the moving platform and where the robot arm will be fixed are (x_0_, y_0_, z_0_).

Two functions are introduced to formulate the objectives:(1)*f_inverse_*(*x*_0_, *y*_0_, *z*_0_) *→* (*Ѳ*_1_, *Ѳ*_2_, *Ѳ*_3_) for solving the inverse kinematic problem(2)*f_forward_*(*Ѳ*_1_*, Ѳ*_2_*, Ѳ*_3_) *→* (*x*_0_, *y*_0_, *z*_0_) for solving the forward kinematic problem.

The solution of these problems plays a key role in the operation and motion control of the delta robot manipulator.

The importance of forward kinematic characterization lies in the processes of trajectory planning, control of the delta robot manipulator, and in the optimization of its motion trajectory to minimize energy and time consumption. This calculation allowed us to calculate the coordinates of the robot’s end based on the rotation angle of its components. 

The inverse kinematic characterization of the delta robot in turn provides a means of calculating the required rotation angles to move the robot to a given position. The importance of this characterization is to ensure that the robot moves accurately to a given position based on its final coordinates, taking into account the correction of the joint rotation angles.

The delta robot comprises two triangular platforms: a fixed platform (1) and a mobile platform (2). These platforms are connected by three kinematic chains, each composed of two links: an active link (2) and a passive link (3). The end effector (4) is affixed to the mobile platform. 

Based on a study by the authors of [5], the forward kinematics of the delta robot manipulator allows the position of the turning elements to be determined based on the known angular rotation. In contrast, inverse kinematics allow the necessary turning positions to be determined to achieve a given robot position element.

### 3.2. Development of the Mechanical Design and Electronic System of the Delta Robot Manipulator

To develop the mechanical design of the Delta Robot Manipulator, a 3D model was generated using SolidWorks 3D CAD Design Software & PDM Systems 2022. The design was carried out considering all technical requirements and functional characteristics necessary for the manipulator’s operation. Detailed drawings of all the components of the manipulator were created using SolidWorks tools, including the frame, motors, links, and other parts.

Additionally, a visualization of the manipulator’s operation was created using SolidWorks, allowing for the checking of its functionality and efficiency. The resulting 3D model, shown in Figure 4, will be used to produce a prototype of the manipulator.

During the development of the Delta Robot Manipulator, the majority of the frame connectors were 3D printed on a Flying Bear Ghost 4S 3D printer using Bestfilament PLA 0.75 mm plastic. The 3D printing process was facilitated by a Cura Ultimaker 3D slicer. The frame elements, as depicted in Figure 5 of the delta robot arm prototype, were modeled in the FreeCad automatic design (3D) software environment. This allowed for the determination of optimal dimensions before being printed using a 3D printer.

In general, the experimental setup consists of the following components of the main and auxiliary devices:-three Nema 17 stepper motors (42HS60-1504-001);-MKS SBase V1.3 control boards;-DRV8825 stepper drivers;-Endstop mechanical switch;-MKS TFT35 V1.0 touch display with a Wi-Fi module.

The upper part of the delta robot, as shown in Figure 6, comprises a flat platform on which manipulators or other working elements, such as the Nema 17 type stepper motors of the brand 42HS60-1504-001, are mounted onto an alloy steel bracket. This bracket is securely fixed to the fixed plate through drilled holes.

A camera with a high color sensitivity is located on a movable plate connected to Raspberry Pi by a cable to take photos or videos of the object located on the bottom surface of the delta robot manipulator.

In this case, the Raspberry Pi 4 (Model B) acts as a centralized controller and processor for processing data received from a color-sensitive camera located on a moving platform. The Raspberry Pi is connected to this camera by a cable and is used to extend the detection, reception, and processing functions of the camera data. After analyzing the received data, the Raspberry Pi commands the motors of the delta robot manipulator, adjusting its movement and positioning to perform various tasks. Additionally, the Raspberry Pi helps interact with other components of the delta robot system to coordinate actions and solve complex tasks, taking into account the storage and analysis of the data.

Also, stepper drivers of the DRV8825 type, as depicted in Figure 7, were used. This type of driver is a device that configures the operating modes of the stepper motors in the delta robot manipulator, controlling parameters such as the angle of rotation and direction of movement.

The wiring diagram of the stepper motor and the control board to the DRV8825 type driver is shown in Figure 8. In this case, it is necessary to ensure the correct connection of the phase wires to guarantee the motor’s proper operation. To achieve this, the phase wires of the stepper motor must be connected to the A1, A2, B1, and B2 outputs on the DRV8825 driver. Additionally, the stepper motor power wire is connected to the VMOT and GND power terminals on the DRV8825 driver. The step (STEP) and direction (DIR) control wires should be connected to the corresponding ports on the MKS SBase V1.3 control board. Furthermore, the power wires (VCC, GND) are connected to the control board, and the input signal wires (EN) are connected to the DRV8825 driver. Finally, the control board is connected to the power supply. After completing the wiring, the motor control program on the MKS SBase V1.3 control board can be initiated.

Special software is utilized to control all components of the system, enabling adjustment of movement parameters of the stepper motors, control of the robot’s positioning, and monitoring of its operation via the touchscreen display. This functionality is facilitated by the proper connection of the delta robot manipulator components in the electronic circuit, as depicted in Figure 9.

The MKS Sbase control board uses Smoothie firmware with the “Rotary delta” configuration file, the beginning of which is shown in Figure 10. This configuration is designed to work with nonlinear delta robots.

### 3.3. Developing an Artificial Vision System for the Delta Robot Manipulator Prototype and Its Machine-to-Machine Communication Protocol

#### 3.3.1. Development of Artificial Vision System for Delta Robot Manipulator Prototype

Based on the analysis of the related work mentioned in [35,36,37,38], in the context of trajectory tracking, the choice between different algorithms depends on the specifics of the task to be performed by the delta robot manipulator. To determine the best algorithm for delta robot manipulator trajectory tracking, it is necessary to analyze the characteristics of the data, the accuracy and performance requirements, and the overall complexity of the task.

CNNs and DNNs (deep neural networks) are commonly used for image analysis and pattern recognition, where the main objective involves analyzing images to track a trajectory in space. These models may be the best choice, as they are able to extract features from images and predict motion based on these features. However, if the system has a small number of training examples, the recognition accuracy of CNNs can decrease significantly. Additionally, DNNs have difficulties in interpreting results and training the model due to the large number of layers and parameters, which require substantial computational resources for training and prediction. This can be problematic in robotic systems where high-speed tasks are required.

Simpler models like basic neural networks (NNs) may be more suitable if the data have a simple structure and do not require deep feature analysis or high accuracy in object detection.

RBF (radial basis function) networks can be effective if the task requires approximation of complex functions or deals with inaccurate information about the data. However, this increases the data processing time.

The artificial vision system for the delta robot manipulator prototype is designed to detect parts during the sorting process. The first step in developing the artificial vision system for the prototype involves collecting images for training, achieved by using image sensors to acquire image data. For prototype manipulators, the most commonly used type of image is color images of the RGB type. Additionally, modern architectures like YOLOv8 are employed in the experimental phase of the study.

The analysis of existing variants of applying machine learning models for the formation of artificial vision systems demonstrates the effectiveness of convolutional neural networks with MASK-R-CNN architecture, where the number of layers varies from 5 to 50. The application of this algorithm is one of the most effective for the process of object segmentation in images, providing the ability to select masks for instances of different objects in photos, even when the objects partially overlap or have different sizes. The structure of the convolutional neural network is presented in Figure 11.

All feature maps in the convolution layer have the same dimensions, which are determined by the following formula:(1)s=mS−kS+1
(2)s’=mS’−kS’+1
where (*s*, *s*’) is the computed size of the convolutional map, *mS* is the width of the previous map, *mS*’ is the height of the previous map, *kS* is the width of the kernel, and *kS*’ is the height of the kernel.

In order to avoid non-calibrated results, the step size can be varied from 0 to 2 relative units when calculating the output feature size of the map.

The application of the MASK-R-CNN neural network architecture for the delta robot manipulator forms the following basic processes of artificial vision elements:-classification;-semantic segmentation;-object detection;-instance segmentation.

Developing an artificial vision system using the MASK-R-CNN method has several advantages over other methods and is an effective tool for creating robotic systems that are capable of efficient real-time image processing.

The rationale behind the application of MASK-R-CNN in this study is as follows:-It provides accurate object detection in images, which can accurately detect object boundaries and create masks that show the contour of each object.-It can process images in real-time, making it an excellent choice for robotics applications where processing speed is important.-It allows for simultaneous object detection, classification, and image segmentation. This makes it a versatile tool for a variety of artificial vision applications.-It is open source, making it easily accessible and easy to use.

Additionally, the application of MASK-R-CNN enables the automation of several tasks related to image processing, thereby increasing the efficiency of robotic systems. The RGB model is an additive model, where visible light is concentrated during the creation of new colors for the image through addition. In this case, new colors are formed by the additive mixing of red, green, and blue in variable proportions, with three stripes in red, green, and blue colors present in these images. In this study, the architecture of the MASK-R-CNN algorithm with an adaptive RGB model is illustrated in Figure 12.

For the delta robot manipulator to recognize objects in RGB-type images, setting up the classification code by object type is sufficient to identify the objects. In this study, the open-source Computer Vision Library was utilized as an open-source computer vision and image processing library. However, for data processing and verification of neural network training results, including the optimization of imaging results, the joint operation of artificial vision based on MASK-R-CNN with machine-to-machine (M2M) protocol is applied. This will contribute to activating the delta robot manipulator into a mobile state. In this study, the implementation of the MASK-R-CNN architecture with an adaptive RGB model was carried out using the selective search algorithm. This algorithm can quickly suggest potential image regions for further object detection based on various characteristics such as color, texture, size, and shape.

The basic idea is to group image pixels into similar regions, which are then combined into larger segments to map objects in the image onto the considered Ozx, Ozy, Oxy delta robot manipulator space for further positioning.

YOLO (you only look once) technology represents an innovative approach to building artificial vision systems that leverage deep learning and computer vision technologies for image and video processing. 

The advantages of using YOLOv8 technology for a delta robot arm include fast data processing and a high-speed algorithm, which allows the robot to react faster to changes in the environment and perform various tasks at high speed. When YOLOv8 technology is integrated with a delta robot arm, functions such as automatic recognition and positioning of objects for further processing or assembly can be realized.

OpenCV (Open Computer Vision Library), which provides powerful tools for image and video processing as well as for the realization of computer vision and machine learning, was used as a dataset. The popularity of its application is confirmed by the implementation of various image operations such as filtering, binarization, segmentation, and object recognition.

In this case, the OpenCV library has been used to improve the functionality of the delta robot manipulator. It allows for the implementation of algorithms to recognize objects in images, determine their position and orientation, and perform automatic control of the arm to grasp and move objects. This enables the robotic arm to move and perform tasks efficiently and accurately.

#### 3.3.2. Integration of Artificial Vision System with Machine-to-Machine Communication Protocol

Integration of the artificial vision system with machine-to-machine communication protocols can significantly expand the functionality of the delta robot manipulator in positioning and orientation within three-dimensional space. The use of these communication protocols, in combination with the artificial vision system, allows data about recognized objects and segmented image areas to be transferred between different devices and systems. This primarily creates a distributed control system that allows the delta robot arm to efficiently navigate and coordinate in space as well as process a large set of real-time data at a high speed for target detection. The configuration of the interconnection between the various components and devices of the delta robot arm is shown in Figure 13.

Here is a simplified block diagram of the delta manipulator system:-Power supply: The power supply provides power to the entire system. It converts AC power from the mains to DC power suitable for other system components.-Stepper controller: The stepper controller is an important part of the system that drives the stepper motors. It interprets the commands of the main controller (for example, a microcontroller or a computer) and generates the necessary signals to control the stepper motors.-Encoder: The encoder provides feedback on the position and speed of the end effector of the manipulator or individual joints. This feedback is important for feedback control and precision positioning tasks. Encoders can be incremental or absolute depending on the requirements of the application.-Stepper gear motor: Stepper gear motors are actuators that are responsible for moving the joints of the manipulator. Stepper motors are preferred in many robotic applications because of their precise control and ability to maintain their position without feedback when properly controlled.

The main controller, which can be a microcontroller or a computer, sends commands to the stepper controller depending on the desired trajectory or tasks. The stepper controller then coordinates the motion of the stepper motors based on these commands and feedback received from the encoders to achieve the desired motion of the delta manipulator end effector.

A machine vision system for identifying and tracking the trajectory of objects usually includes several components working together to collect, process, and analyze visual data. 

-Camera: The camera captures images or video footage of a scene where objects are present. The type of camera (RGB) depends on the specific requirements of the application, such as the need for color information or depth perception.-Lens: The lens focuses light on the camera matrix and determines aspects such as field of view, depth of field, and focal length. Depending on the specific requirements of the application, such as the distance from the tracked objects and the required level of detail, different lenses can be used.

Lighting: Proper lighting is crucial for good image quality and accurate object detection. Various lighting methods can be used, including ambient lighting, LED arrays, or strobe lights, depending on factors such as ambient lighting conditions and reflective properties of objects.

-Image processing unit: This unit processes images captured by the camera to extract relevant information about objects in the scene. Image processing techniques can include noise reduction, image enhancement, segmentation (to separate objects from the background), feature extraction, and pattern recognition.-An algorithm for detecting and tracking objects: Object detection algorithms identify and detect objects inside images or video frames. These algorithms can use techniques such as pattern matching, edge detection, contour detection, or machine learning-based approaches such as convolutional neural networks (MASK-R-CNN) for more complex scenarios. After objects are detected, tracking algorithms predict and update the positions of objects in successive frames, providing trajectory tracking.-Trajectory analysis and prediction: Trajectory analysis algorithms analyze patterns of movement of objects over time to predict their future positions and trajectories. These algorithms can use techniques such as Kalman filters, particle filters, or optical flow analysis to estimate the velocity, acceleration, and direction of movement of an object.-Feedback and management system: Trajectory information obtained from the vision system can be used to provide feedback to a control system or a robotic platform for real-time decision-making or adjustments. For example, in a robotic arm, trajectory information can be used to adjust the position and orientation of the arm to accurately interact with moving objects.

By integrating these components, a machine vision system can effectively identify and track the trajectory of objects. Figure 14 shows the scheme of machine vision for identifying and tracking the trajectory of objects.

In the conducted study, a highly color-sensitive camera was used to capture images of objects, with subsequent processing using software applications. The captured object data were then transferred to Raspberry Pi. The Raspberry Pi performed data processing using Python programming language scripts and the OpenCV library for object detection. The processing results were then sent to the Arduino (type of Mega) through the serial port. The Arduino, in turn, controlled the motors to move the moving mechanisms of the manipulator based on the data received. The serial communication between the Raspberry Pi and Arduino provided bidirectional data transfer, allowing both devices to communicate with each other. In a typical setup, the Arduino could transmit data to the Raspberry Pi and also receive data from it and vice versa.

The Raspberry Pi acts as the central hardware entity that processes data from the camera and controls all the working systems. It performs image analysis using Python scripts and the OpenCV library to search for objects. It also processes data, taking images from the camera, analyzing them to find objects, and determining the necessary actions. Based on the results of image processing, it sends commands to the Arduino to control the manipulator motors. The Arduino is responsible for directly controlling the delta robot manipulator motors. It receives commands from the Raspberry Pi through a serial port and provides the physical manipulator movement according to the submitted tasks. Additionally, it directly interacts with the mechanical parts of the manipulator, including the drive motor and moving mechanisms.

Through the serial port, the Arduino can both receive commands from the Raspberry Pi and send back information about the state of the manipulator, allowing the system to respond to changes in real-time. This combination creates a flexible and powerful system in which the Raspberry Pi handles the complex tasks of image analysis and logic control for the delta robot arm, while the Arduino provides direct and efficient control of the physical components of the arm. A block diagram of machine-to-machine interaction (M2M) for artificial vision is illustrated in Figure 15.

The robotic platform allows for predicting of the trajectory of an object based on input data and outputting the results in a user-friendly format. The scheme of operation of the robotic platform for predicting the trajectory of movement is illustrated in Figure 16.

Integration of the artificial vision system with the delta robot manipulator control system is performed through machine-to-machine communication, which is realized through an ethernet interface designed for data exchange and processing between different systems. The scheme of operation of the delta manipulator control model is illustrated in Figure 17.

The main components and their interactions according to the control scheme shown in Figure 17 are as follows:

*Inverse Kinematics Model*: Receives data from the sensors about the robot’s position and orientation and calculates the necessary angles for the stepper motors to achieve the given position and orientation.

*Stepper Motors*: Drive the manipulator according to the calculated angles from the inverse kinematics model.

*Robot Pose:* Reflects the current position and orientation of the manipulator.

*Position Ref*(*t*) *and Attitude Ref*(*t*): Define the setpoints for the manipulator’s position and orientation to be achieved.

*Error Signals* (*e*(*t*)): Determine the difference between the setpoints (Position Ref(t) and Attitude Ref(t)) and the current manipulator position and orientation values.

*PI Controllers*: Consist of proportional (KP, which responds to the current error) and integral (KI, which accounts for accumulated errors over time, helping to eliminate steady-state errors) components, generating control signals (u(t)) to correct the error.

*Control Signals* (*u*(*t*)): The output signals of the PI controllers that adjust the operation of the stepper motors to minimize errors and achieve the position and orientation setpoints.

Although PI control is a powerful method for controlling robots, it can encounter difficulties when the Ref(t) task changes over time, especially if the changes are rapid or unpredictable. To improve the control system, it may be necessary to add a differential component or to use adaptive and predictive control methods. Additionally, the control accuracy of the manipulator is highly dependent on the correctness of the inverse kinematics model; any inaccuracies in the model can lead to errors in the position and orientation of the manipulator. In this study, to avoid these problems, the PI controllers are precisely tuned and optimized for specific manipulator operating conditions to ensure stable operation.

This scheme of operation of the delta manipulator control model can be realized with the help of software and a control room that enables the control of the movement of each link, monitors their position and orientation, and corrects the movement to reach the desired end point. 

The proposed structure setup of the developed artificial vision system using M2M protocols provides a structured and efficient transfer of information between delta robot-manipulator devices with a high accuracy in detecting objects and concentrated positioning towards the detected target.

## 4. Experimental Study of the Delta Robot Manipulator Prototype

The assembled fully functional delta robot manipulator illustrated in Figure 18 can find application in positioning tasks and the processes of sorting and assembling parts, where the speed of manipulation and accuracy of object detection are important. For this purpose, an artificial vision system for the delta robot manipulator was developed, which is a rather complex task requiring the integration of various systems and components.

To obtain an objective evaluation of the performance of an M2M-based artificial vision system for a robot manipulator in positioning tasks, an experimental study was conducted in a standardized work environment that closely resembles a production environment. This study took into account the most important parameters that affect the target detection process, including lighting, background, and the size and location of objects to be positioned, in order to eliminate the influence of external factors on the results. The total area of the room was 46 m^2^, and the distance between the upper and lower platforms was 1 m (100 cm).

The experiments were conducted at a normal temperature of approximately 25 °C, which primarily allowed the baseline performance of the system to be evaluated under standard conditions.

To experiment with a machine learning-based artificial vision (M2M) system for a robot manipulator in a positioning task, it is important to consider different lighting levels, including standard daylight, artificial lighting, and low light and shadow conditions.

Standard daylight: 5000 lux. This lighting provides good visibility and contrast, which is important for accurate object recognition.

Artificial lighting: 2000 lux. Artificial lighting is used to create uniform illumination of surfaces and objects.

Low light and shadows: 1500 lux. Under these conditions, lighting is insufficient to see objects clearly, which can lead to recognition and positioning problems.

When planning an experiment, it is important to consider that each type of illumination can affect the quality of images produced by the artificial vision system and the ability of the robot manipulator to accurately position objects.

The developed artificial vision system for the delta manipulator robot should fulfill the following tasks:-Object detection and identification, selection of areas in the considered space;-Positioning of the delta robot manipulator in space, planning of movement trajectories, and interaction with objects.

To realize the first stage of the task, an artificial vision system was designed for the delta robot manipulator prototype. In addition, interaction with the machine-to-machine communication protocol was applied for positioning and determining motion trajectories for object detection.

The software part of this study was implemented using a joint combination of C++23 and Python 3.9 programming languages. By introducing the coordinates of the forward and inverse kinematics of the delta robot manipulator, a special program code that describes the algorithm of the delta robot manipulator was written in the C++ programming environment, and the program code for working with the artificial vision system was implemented in the Python programming environment. By applying the methods of object detection and recognition, the main tasks of the computer vision system were solved, namely object detection, focusing on it, and tracking. The main parameters and properties of the computer vision system algorithm were also written in the Python environment.

In this experimental study, the initial step of the delta robot arm operation is to determine the initial coordinates of the lower base of the manipulator in three-dimensional space. This is illustrated in Figure 19a, which shows the delta robot arm in the operating state, with the main components connected. The delta robot arm then changes its position in Figure 19b to determine the initial coordinates of the lower base and switches to the waiting mode of pointing towards the object in the visible view of the manipulator.

Once the objects in the delta robot arm’s view have been detected and segmented, LAB color space recognition methods can be used to filter the objects based on their color. For this purpose, in the experimental study, position transformation from the camera frame to the delta robot arm frame in LAB color space is performed. The application of the LAB color space recognition method with the combination of the MASK-R-CNN algorithm improves the accuracy and efficiency of object detection in images, especially when color is an important feature for their identification or classification.

The presented software interface in Figure 20 consists of a code terminal (code algorithm for object detection and recognition), a serial terminal (reading output parameters and results), a frame buffer (dynamic video stream, 1fps), and histograms. In the experimental study, in the LAB color space, the luminance value is placed separately from the tone and color saturation values. Lightness in this case is given by the coordinate L, which varies on a scale from 0 to 100; that is, from the level of the darkest to the lightest tone, the chromatic component is formed using two Cartesian coordinates, A and B. In the experimental study, the first (A) denotes the color position in the range from green to red, and the second (B) denotes the range from blue to yellow.

This software, with a special program code, makes it possible to realize effective work of the computer vision system in real-time mode due to the presence of a “Frame Buffer” block, with the further translation of dynamic video streams in real-time. Also, this software has a block, “Histogram”, which contains dynamically time-varying parametric graphs. The said graphs in the interface determine the color space values when objects are detected. The color space values determine and set the properties of the parameters of the object under investigation using the LAB method.

## 5. Results and Discussion

Unlike the color spaces of various types of cameras, which essentially represent a set of hardware data for reproducing color on paper or a monitor screen (where color may vary depending on factors such as the type of printing press, brand of ink, humidity in the shop, or monitor manufacturer and its settings), a delta-robot manipulator frame based on LAB defines a color space with a high precision. Therefore, LAB is widely used in image processing software as an intermediate color space through which data conversion between other color spaces (e.g., from RGB scanner to CMYK) takes place. Additionally, the special properties of LAB make it a powerful color correction tool.

Next, object detection and positioning experiments were conducted to investigate the performance of the developed artificial vision system with M2M. During the implementation of the experimental part of this study, a comparative analysis of the application of the MASK-R-CNN algorithm architecture with the adaptive RGB and YOLOv8 models was conducted. 

The object of study as a target for detection in the experimental part is a red-colored circular shape with a diameter of 41 mm (red, circle) and a green-colored square of 48 mm^2^ (green, square).

By properly positioning the delta manipulator of the robot depending on changes in the location of the sensing object, a high accuracy and efficiency of the task can be achieved.

As shown in Figure 21, the robot reacts to changes in the environment and moves towards the object to obtain a better view or pick it up. This is facilitated by M2M sensors and artificial vision algorithms, which enable the robot to adapt to changes and make decisions based on the received information. Thus, it can adjust its position to perform a specific task by moving the movable part of the arm towards the object’s location. 

Figure 22 and Figure 23 show that training the MASK-R-CNN model using YOLOv8 for 50 epochs resulted in the following results on the test sample of the experimental study. This confirms the potential of the MASK-R-CNN model with YOLOv8 in object detection for delta robot manipulator positioning, but further work is required to improve its performance for more real-world applications.

The results indicate that the MASK-R-CNN model using YOLOv8 is quite successful in recognizing “red, circle” (Figure 21) and “green, square” (Figure 22) objects. The recognition accuracy is close to 80%, which is a good result. The object detection time is also quite acceptable for real-time use.

One of the main drawbacks of the YOLOv8 model (and the previous version of YOLO) is its sensitivity to changes in brightness and color illumination. YOLOv8, like other convolutional neural network (CNN)-based models, is trained on static images that may differ from the conditions in which the model will be used. As a result, the model’s accuracy in object detection can significantly decrease if the illumination level or colored background is different. Further improvements to the model can be achieved by adding more training data, improving the model architecture, or using other methods to improve the accuracy and speed of object detection.

The results of applying the MASK-R-CNN algorithm architecture with the adaptive RGB model are shown in Figure 24 and Figure 25, where the object is to be recognized as a red-colored circular shape with a diameter of 41 mm (red, circle) and a green-colored square of 48 mm^2^ (green, square).

The nature of color detection in the LAB color space using the original image provided an opportunity to separately affect the brightness and contrast of the image and its color. In the experiment, it allowed us to accelerate image processing for further target detection and provided an opportunity to speed up the process of changing the manipulator position depending on the change in the object location.

The adapted MASK-R-CNN model using RGB data has the potential for use in a variety of applications but requires further optimization work to ensure acceptable real-time performance.

In this experiment, the MASK-R-CNN model with the adaptive RGB model demonstrates a high object recognition accuracy. For both objects, “red, circle” and “green, square”, significant accuracy rates of 0.905 and 0.943 are achieved, respectively. This indicates the effectiveness of the adapted model in recognizing objects. The time taken to detect each object was about 2.000 s. This is quite a long time and requires further optimization of the model to reduce the processing time. Table 1 shows a comparison of the results of the MASK-R-CNN algorithm models.

The main disadvantage of the YOLOv8 model can be related to its complexity and computational resource intensity, which makes it less suitable for application on devices with limited resources or in tasks that require high real-time processing speeds, such as delta robot manipulators.

The application of the LAB method based on the MASK-R-CNN algorithm architecture with an adaptive RGB model is more suitable for the task of a computer vision system related to the detection and recognition of the investigated object. The result of using this method was a clearer definition of the boundaries of the color space of the object under study, which effectively increased the efficiency of the computer vision system tasks. Additionally, the inter-machine interconnectivity simplified the process of controlling the delta-robot manipulator, allowing for rapid movement of the manipulator towards the detected object.

The period of data processing in the experiment was not more than 1 s, but due to the technical features of the modules used and the protocols of machine-to-machine communication, the detection process varied from 1 to 2 s.

In the recognition accuracy criterion, MASK-R-CNN with the adaptive RGB model provides a higher recognition accuracy under both 2000 lux and 1500 lux illumination conditions compared to YOLOv8, according to the results shown in Table 2 and Table 3.

In terms of the detection time criterion, YOLOv8 has a faster object detection time in both illumination conditions, even though its accuracy is lower.

The choice between MASK-R-CNN and YOLOv8 depends on the priorities of the task. If a high recognition accuracy is required, MASK-R-CNN is the preferred choice. If the speed of object detection is more important, YOLOv8 is a better choice.

MASK-R-CNN is well-suited for object segmentation in images and has a higher accuracy. It can accurately detect object boundaries and contours, which is useful for tasks requiring very accurate object recognition. On the other hand, YOLOv8 is known for its fast performance and can detect objects in images quickly. It is well-suited for problems where the speed of object detection is important.

Thus, the choice between MASK-R-CNN and YOLOv8 should be based on the requirements of the task performed by the delta robot, prioritizing the importance of accuracy and speed criteria.

The application of the LAB method, based on the MASK-R-CNN algorithm architecture with the adaptive RGB model, is more suitable for solving computer vision tasks related to the detection and recognition of the object under study. The result of using this method was a clearer definition of the boundaries of the color space of the object under study, which effectively improved the efficiency of computer vision system tasks. Additionally, the machine-to-machine interconnection simplified the process of controlling the delta robot manipulator, allowing the manipulator to move quickly and precisely toward the detected object.

## 6. Conclusions

During the first stage, the kinematics equations were derived for the construction of the manipulator, taking into account its geometrical parameters. This made it possible to determine the position of the lower end relative to the basic units of the manipulator.

In the second stage, the mechanical design of the manipulator was carried out, including material selection and strength and stiffness calculations of the parts. The electronic control system of the manipulator was also designed, including the selection of controllers, motors, and sensors, which contributed to the creation of the manipulator control system.

The third stage of the work involved the creation of an artificial vision system for the manipulator through which it could interact with the environment. This system took into account recognizing objects and coordinating the movements of the manipulator using a machine-to-machine communication protocol.

In the final stage, an experimental study of a fully functional delta robot manipulator was carried out. Its characteristics, performance, and accuracy of object detection and positioning were tested. The results obtained helped us to determine the effectiveness and potential of using this manipulator in various industrial and robotics applications.

In this experimental study, the nature of the delta robot manipulator workflow was concluded as follows:

By applying the artificial vision algorithm MASK-R-CNN, the delta robot manipulator system performs the detection and classification of objects in the visible three-dimensional space. It then transmits the data to carry out the segmentation process, where the image is divided into individual segments, ultimately allowing for more accurate identification of the object and its distribution contour in space. After segmentation, the position and orientation of the object of study are determined. In this case, the image detection analysis determines the exact position and orientation of the detected object.

In this experimental study, the processing of received data and the transfer of visual data between different components of the system were carried out using the ethernet M2M protocol. This protocol enabled the effective implementation of manipulator control commands based on the received data from the artificial vision system and facilitated receiving feedback on the performed operations. This, in turn, allowed for optimal generation of motion planning trajectories. In other words, the system plans optimal motion trajectories for the manipulator based on data on the object’s location.

The developed system of artificial vision with the interaction of machine-to-machine communication protocols for the delta robot manipulator in this study meets the above requirements and provides a high accuracy of object detection, with further motion planning and control of the manipulator based on the processing of the received visual information.

Such a system can be used in various industrial and automated scenarios, such as sorting objects on a conveyor, controlling the assembly or packaging process, and other real-time object manipulation tasks.

## Figures and Tables

**Figure 1 sensors-24-03792-f001:**
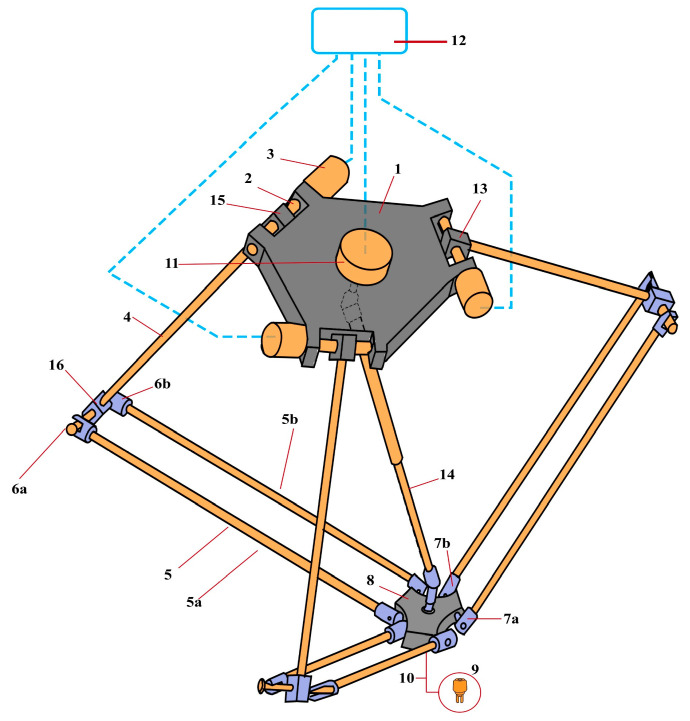
Delta robot manipulator design model.

**Figure 2 sensors-24-03792-f002:**
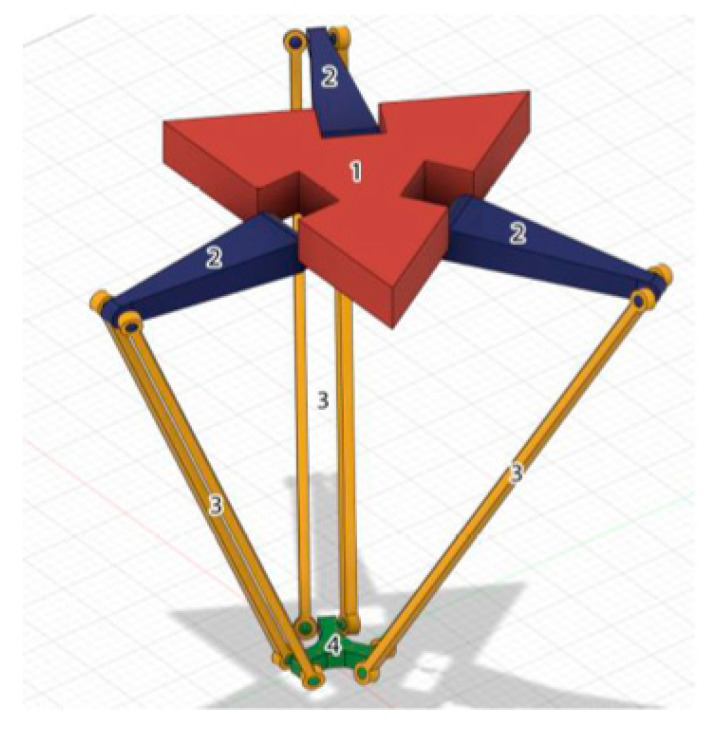
3D model of the delta robot [5].

**Figure 3 sensors-24-03792-f003:**
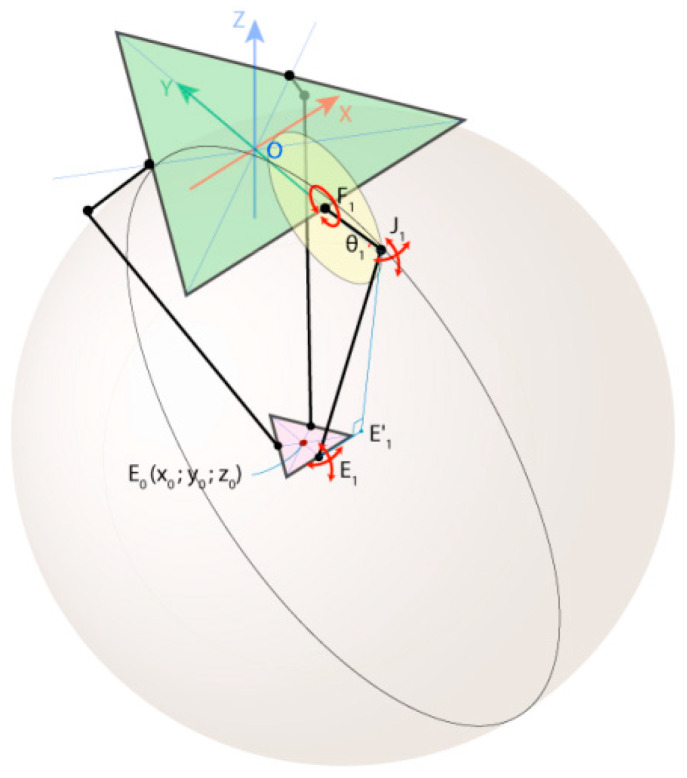
Key parameters of the forward and inverse kinematic problem of the delta robot manipulator.

**Figure 4 sensors-24-03792-f004:**
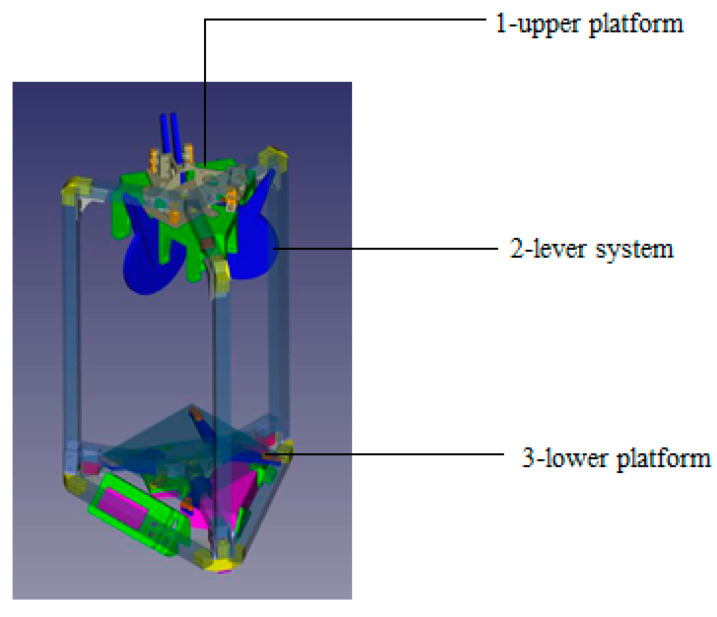
3D model of the delta robot (general view).

**Figure 5 sensors-24-03792-f005:**
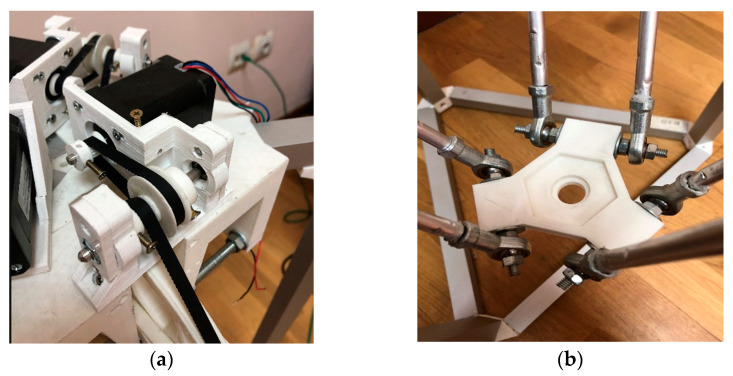
Elements of the upper and lower platform. (**a**) “Upper carrier” element of the upper platform to support the gearbelt system. (**b**) “Carriage” element of the lower platform of the delta robot.

**Figure 6 sensors-24-03792-f006:**
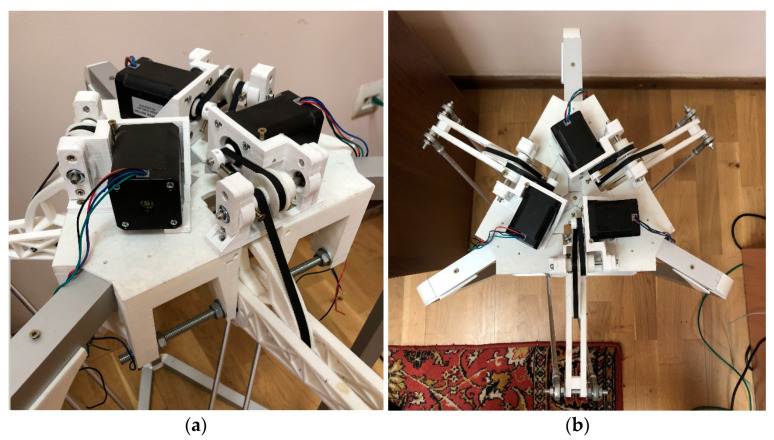
Delta robot manipulator top platform. (**a**) “Top Holder” element of the upper platform with the gearbelt system. (**b**) General view from the top of the upper platform.

**Figure 7 sensors-24-03792-f007:**
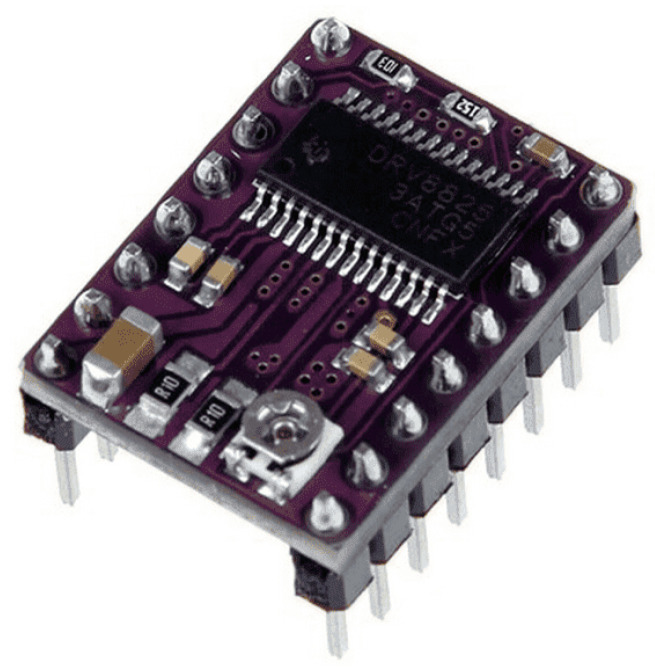
Driver type DRV8825.

**Figure 8 sensors-24-03792-f008:**
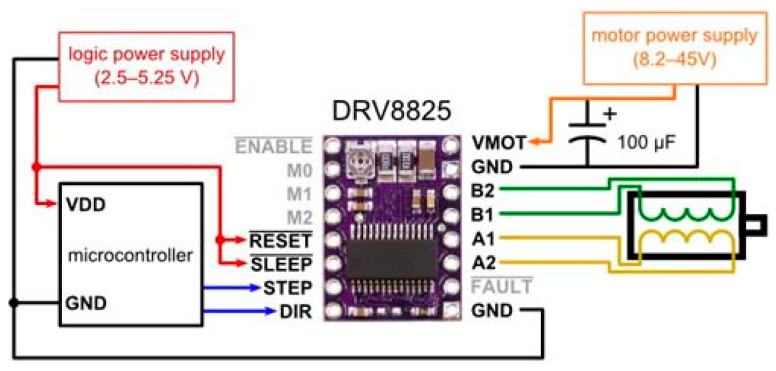
Electronic wiring diagram of DRV8825 driver.

**Figure 9 sensors-24-03792-f009:**
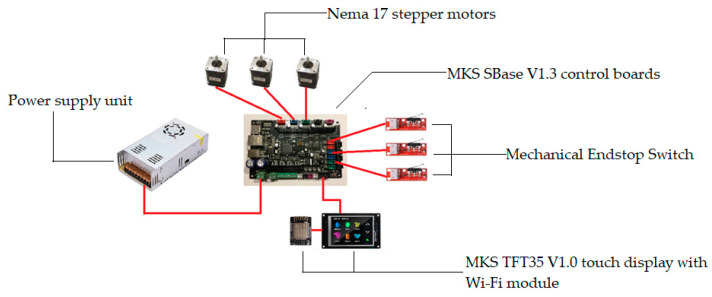
Electronic wiring diagram of delta robot manipulator components.

**Figure 10 sensors-24-03792-f010:**
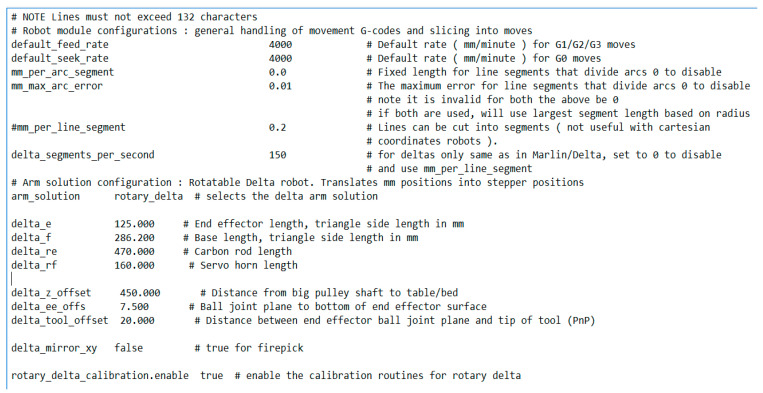
Configuring the “Rotary delta” config file.

**Figure 11 sensors-24-03792-f011:**
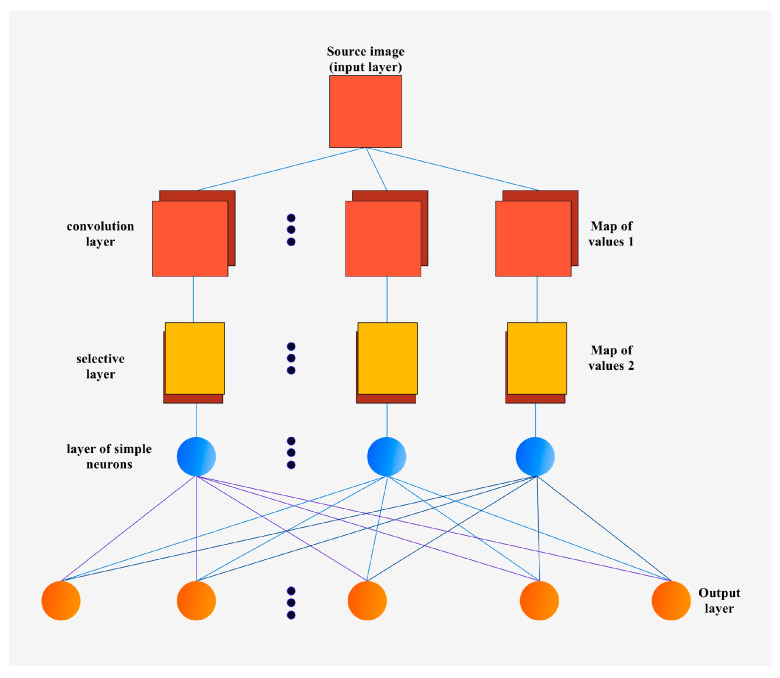
Structure of convolutional neural network.

**Figure 12 sensors-24-03792-f012:**
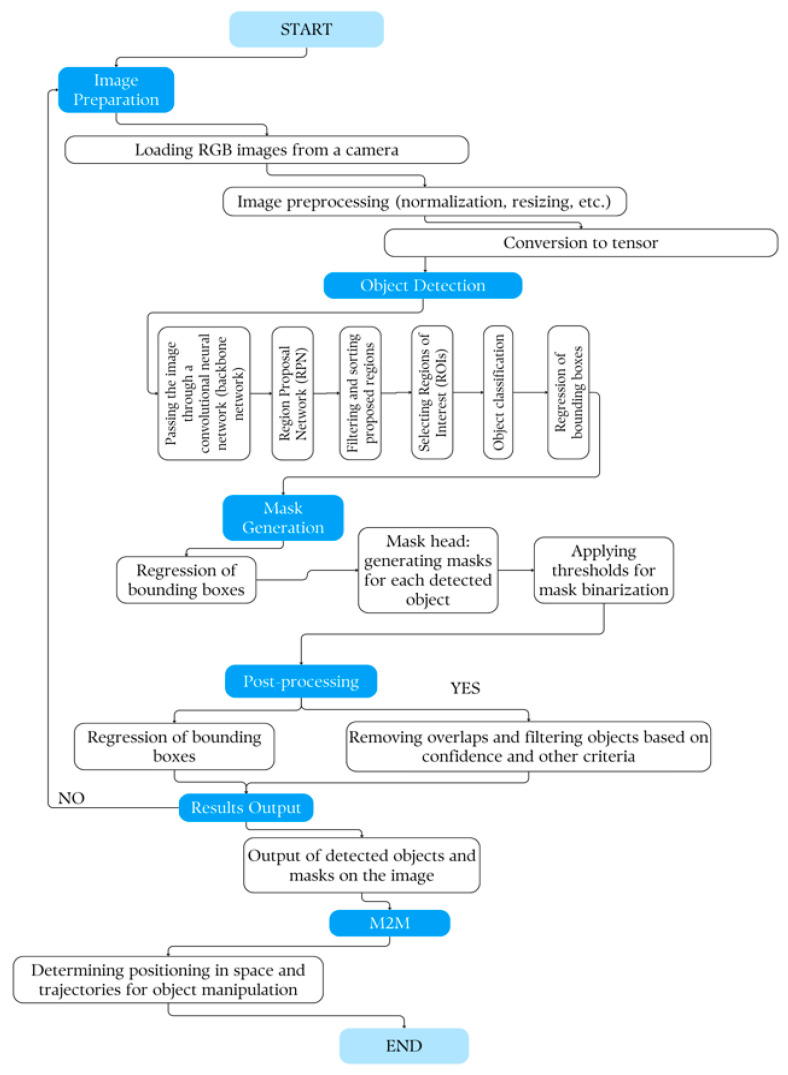
Architecture of MASK-R-CNN algorithm with adaptive RGB model.

**Figure 13 sensors-24-03792-f013:**
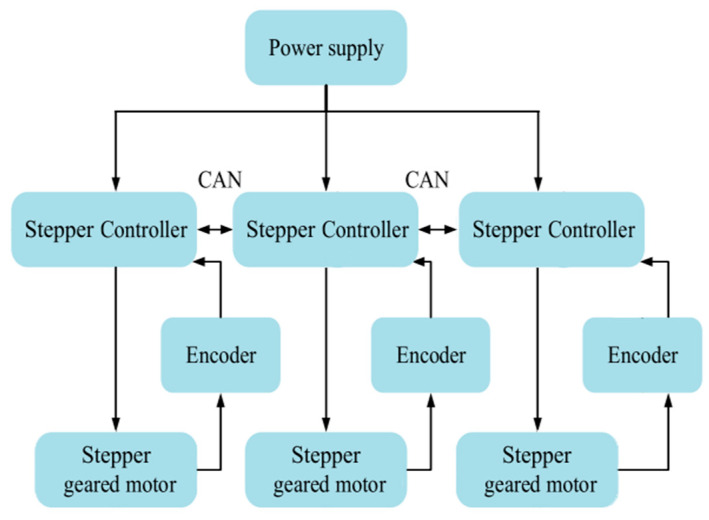
Block diagram of the delta manipulator.

**Figure 14 sensors-24-03792-f014:**
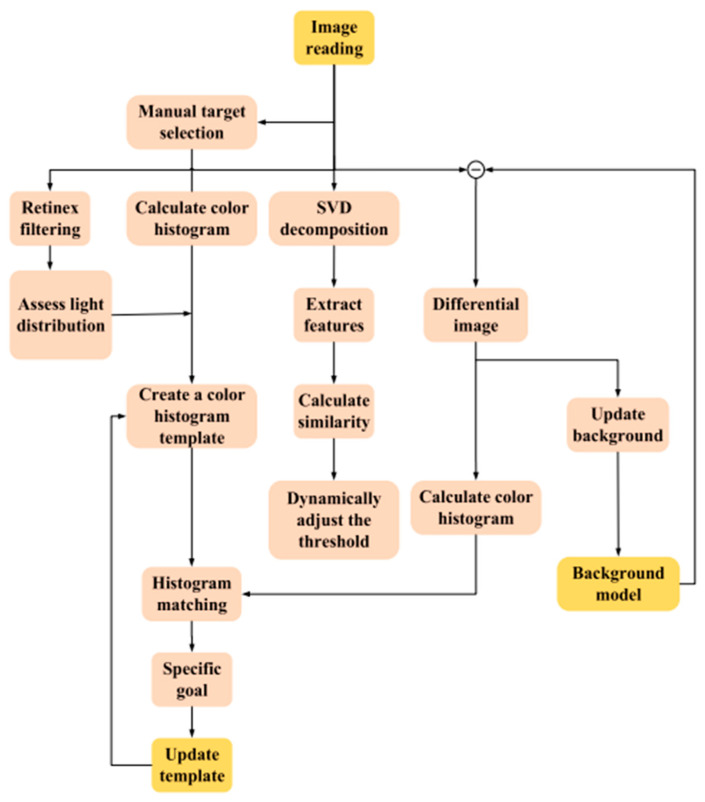
The scheme of machine vision for identifying and tracking the trajectory of objects.

**Figure 15 sensors-24-03792-f015:**
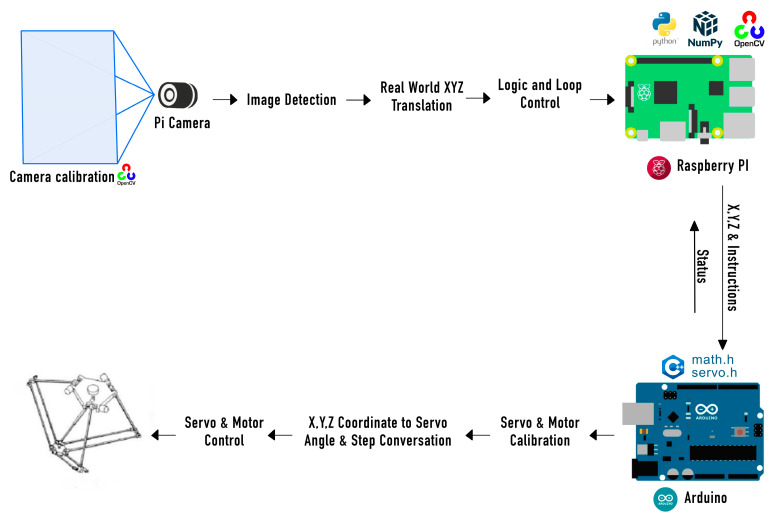
Block diagram of machine-to-machine interaction (M2M) for artificial vision.

**Figure 16 sensors-24-03792-f016:**
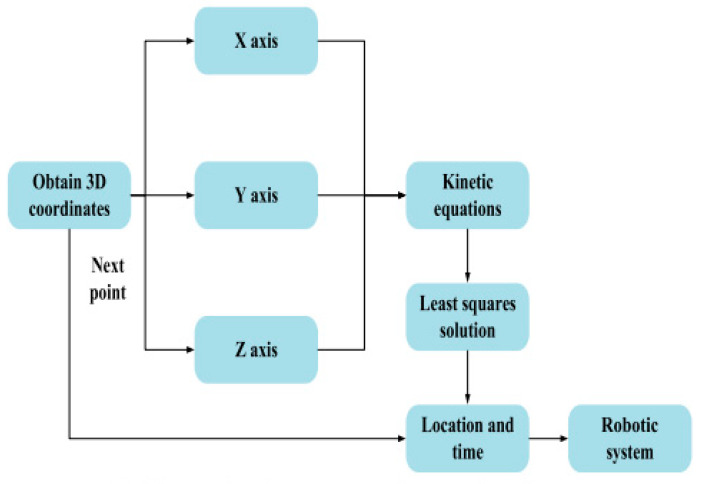
Scheme of operation of the robotic platform for predicting the trajectory of movement.

**Figure 17 sensors-24-03792-f017:**
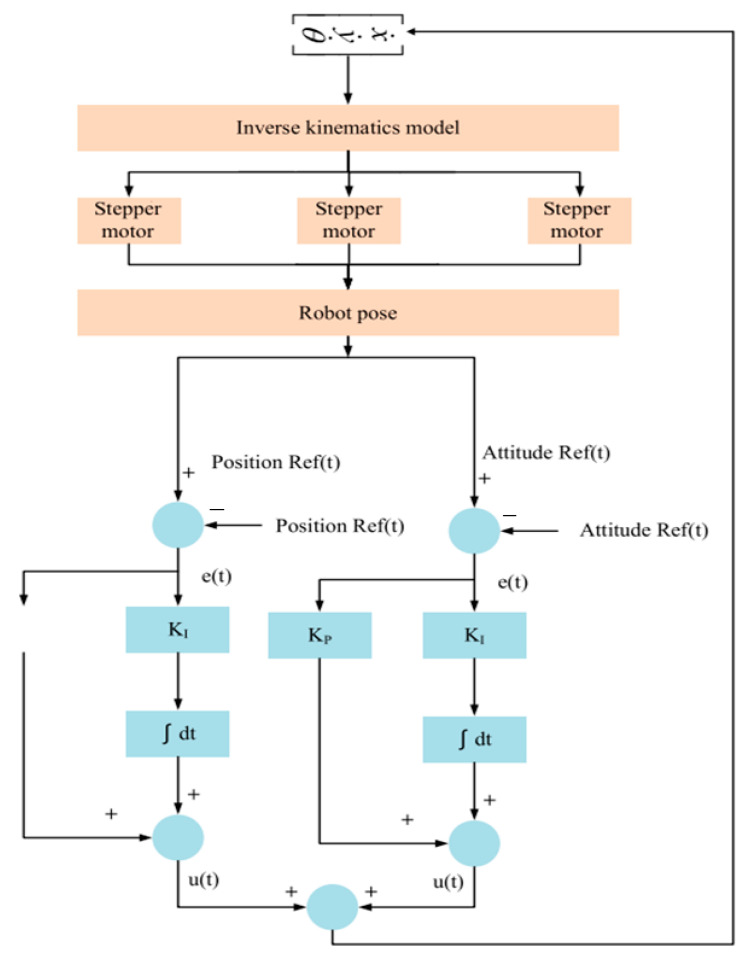
Scheme of operation of the delta manipulator control model.

**Figure 18 sensors-24-03792-f018:**
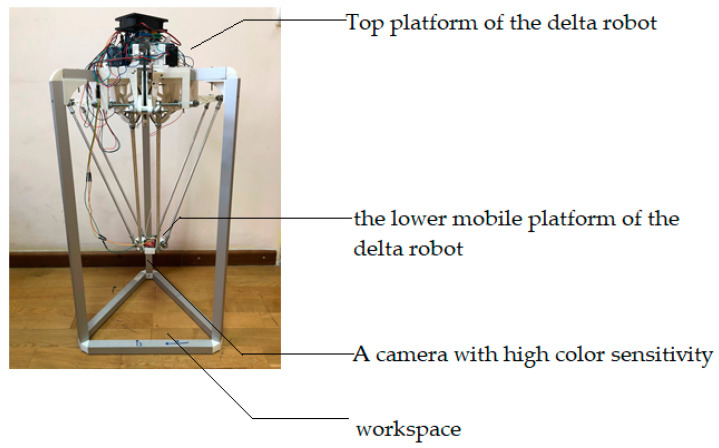
The initial position of the delta robot manipulator.

**Figure 19 sensors-24-03792-f019:**
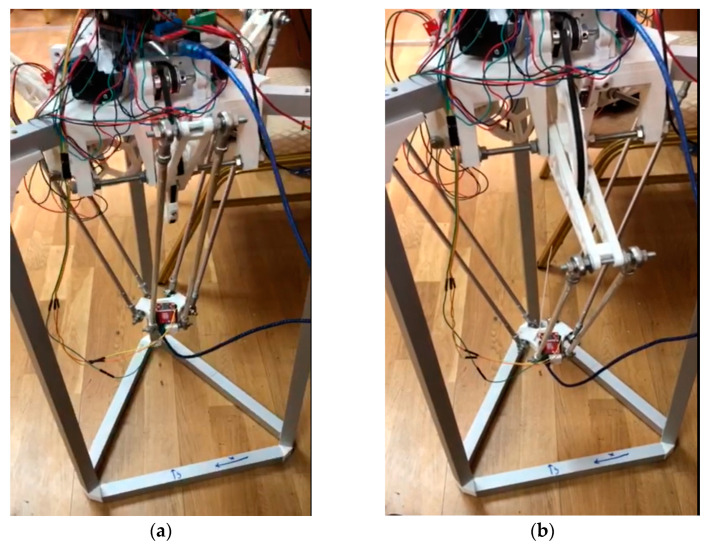
Operating position of the delta robot arm. (**a**) Initial position of the delta robot arm at startup. (**b**) Standby position after the initial coordinates of the lower base have been determined.

**Figure 20 sensors-24-03792-f020:**
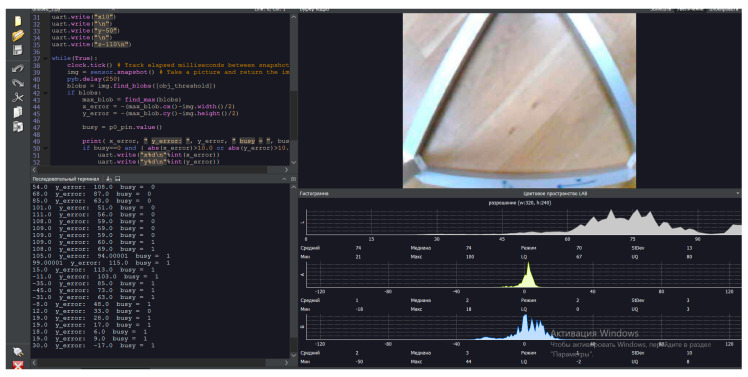
Python software interface when realizing the process of determining the initial coordinates of the bottom base.

**Figure 21 sensors-24-03792-f021:**
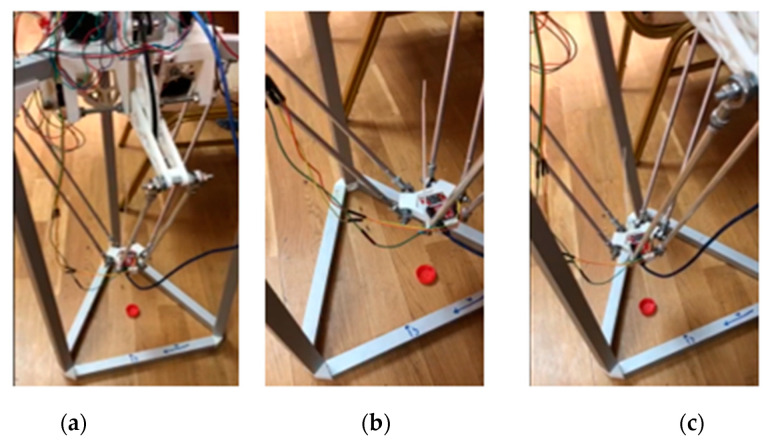
Positioning of the delta robot arm depending on changes in the location of the sensing object. (**a**) The object under study is located in the center, and the manipulator position is adapted to the side of the object, towards the center. (**b**) When the object’s location changes to the right side, the manipulator position is shifted to the right side of the workspace. (**c**) When the object’s location changes to the left side, the manipulator position is shifted to the left side of the workspace.

**Figure 22 sensors-24-03792-f022:**
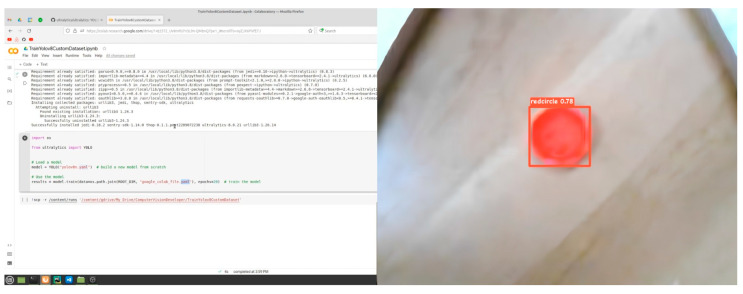
Software interface of detection and centered focusing processes over an object with the parameters of color and shape as “red, circle” with the application of YOLOv8 architecture.

**Figure 23 sensors-24-03792-f023:**
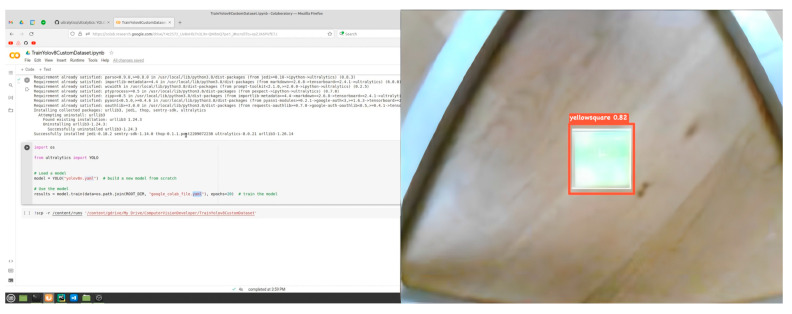
Detection and centered focusing over an object with the parameters of color and shape as “green, square” using YOLOv8 architecture.

**Figure 24 sensors-24-03792-f024:**
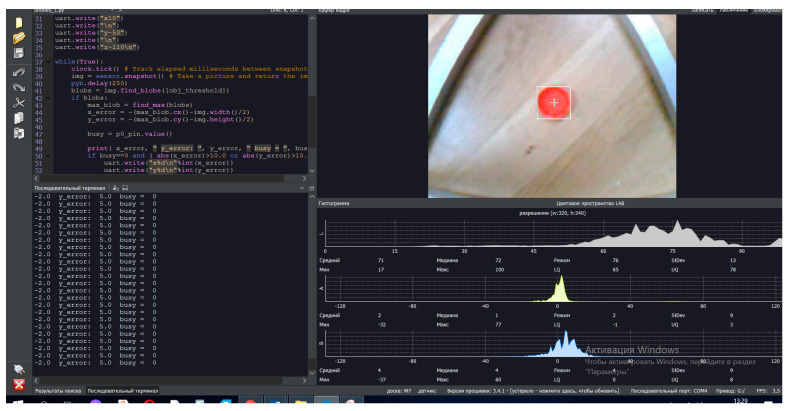
The software interface of detection and centered focusing processes over the object with parameters of color and shape as “red, circle” with the application of the MASK-R-CNN algorithm architecture with an adaptive RGB model.

**Figure 25 sensors-24-03792-f025:**
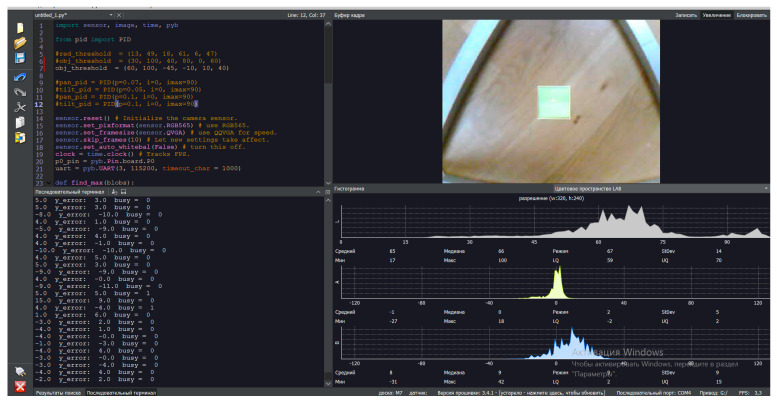
Detection and centered focusing over an object with parameters of color and shape as “green, square” using the MASK-R-CNN algorithm architecture with an adaptive RGB model.

**Table 1 sensors-24-03792-t001:** Comparison of MASK-R-CNN algorithm models under normal environmental conditions (temperature 25 °C, illumination 5000 lux).

No.	Model	Number of Learning Epochs	Recognition Accuracy on the Test Sample		Object Detection Time, sec	
			Red, Circle	Green, Square	Red, Circle	Green, Square
1	YOLOv8	50	0.78	0.82	1.497	1.568
2	RGB	50	0.905	0.943	2.001	1.965

**Table 2 sensors-24-03792-t002:** Comparison of MASK-R-CNN algorithm models under normal environmental conditions (temperature 25 °C, illumination 2000 lux).

No.	Model	Number of Learning Epochs	Recognition Accuracy on the Test Sample		Object Detection Time, sec	
			Red, Circle	Green, Square	Red, Circle	Green, Square
1	YOLOv8	50	0.696	0.745	2.001	2.743
2	RGB	50	0.808	0.785	2.404	2.750

**Table 3 sensors-24-03792-t003:** Comparison of MASK-R-CNN algorithm models under normal environmental conditions (temperature 25 °C, illumination 1500 lux).

No.	Model	Number of Learning Epochs	Recognition Accuracy on the Test Sample		Object Detection Time, sec	
			Red, Circle	Green, Square	Red, Circle	Green, Square
1	YOLOv8	50	0.59	0.709	2.5	2.890
2	RGB	50	0.789	0.763	2.7	2.98

## Data Availability

The data generated in this study are presented in the article. For any clarifications, please contact the corresponding author.

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
