# Peer review of "Development of an Artificial Vision for a Parallel Manipulator Using Machine-to-Machine Technologies"

_sensors, 2024, doi:10.3390/s24123792_

Round 1
Reviewer 1 Report
Comments and Suggestions for Authors
In this article, the author has developed an artificial vision system for a flexible Delta robotic manipulator and integrated it with Machine-to-Machine (M2M) communication technology to optimize real-time device interaction. This integration is designed to enhance the speed and overall performance of the robotic system. Specifically, an artificial vision system for a Delta robotic manipulator was developed, utilizing M2M technology for operation, with RGB images serving as input data. The MASK-RCNN algorithm was employed for processing, and the results were tailored to the characteristics of the Delta robot prototype. The article provides a comprehensive overview of the design and research of the Delta robotic manipulator prototype, addressing kinematic issues in the manipulator's construction, developing mechanical designs and electronic systems, creating an artificial vision system and its corresponding M2M communication protocol, and conducting experimental studies of the artificial vision system under practical working conditions. The content of the article is well-structured, the thought process is clear, and there are innovative elements in parts of the content. However, I would like to include some comments on this work to improve certain aspects.
1. The article provides a sufficient background introduction to the current development of artificial intelligence and robotic systems. However, it is recommended that the authors more clearly articulate how this study differs from existing work and how it fills gaps in existing research.
2. The experimental section clearly demonstrates the performance of the system. The authors are recommended to provide more information about the experimental setup, including test conditions, details of the dataset used, and hyperparameter settings.
3. For the discussion of the results, it is recommended that the authors further explore the performance limitations of the model and its robustness under different lighting conditions or complex backgrounds.
4. The article mentions the use of Raspberry Pi and Arduino for system integration. The authors are recommended to provide more details on hardware selection, software architecture, and system integration.
5. The results section clearly displays the experimental data and is recommended to be further enhanced by adding more qualitative and quantitative experimental analysis charts.
The comparison between MASK-RCNN and YOLOv8 models is mentioned in the article, but insufficient comparative analysis is provided to show the advantages and disadvantages of these two methods. The authors are recommended to detail the performance of each method in specific application scenarios and discuss why MASK-RCNN was chosen as the final solution.
Comments on the Quality of English LanguageMinor editing of English language is recommended.
Author Response
Thank you very much for taking the time to review this manuscript. Please find the detailed responses below and the corresponding corrections highlighted changes in the re-submitted files. Please see the attachment.

Reviewer 2 Report
Comments and Suggestions for Authors
This article discussed the development of an artificial Vision for a Parallel Manipulator using M2M technologies. Although this work describes many related contents (mechanical systems, electronics device, motion identification, ...) and the verification was given by experiment, but this work lacks the deep analysis. Some comments need to be clarified as:
1. Although this submission considers 7 problems (see Page 5) with Problems 3, 4, 6, 7 mentioned the Control System, but the control design has not been analyzed in this submission. Hence, please give some control analyses in the revision;
2. A number of 22 references is is too little in a scientific publication and two references 19, 22 are out of date. Please add more appropriate recent references;
3. Please analyze the satisfaction of Control Diagram in Fig. 17 with PID controller. It should be noted that PID control is not guaranteed the tracking problem in case of time varying reference Ref (t).
4. The Neural Network (NN) was mentioned in this submission with Convolutional NN (Fig. 10). However, many easier NN structure has been employed to apply in trajectory tracking control, such RBF (Radial Basis function network) in https://www.sciencedirect.com/science/article/abs/pii/S0019057822001495, https://ieeexplore.ieee.org/abstract/document/10171231, https://onlinelibrary.wiley.com/doi/abs/10.1002/rnc.6597, https://onlinelibrary.wiley.com/doi/abs/10.1002/rnc.7083, etc. Please give some comparisons to highlight the contribution.
Comments on the Quality of English LanguageSome English sentences need to be corrected as:
1. In line 474, "the functionality of the delta robot - -";
2. In line 685, "This is because YOLOv8, ..."
3. In line 716, 717, "This is quite a long time and can be improved and requires further optimization ..."
4. The title "Development an Artificial ?"
Author Response

(The authors gave the same response as above.)

Reviewer 3 Report
Comments and Suggestions for Authors
M2M interaction based on AI technology is an important and pressing task for researchers around the world and the topic of the manuscript is relevant for the field, but have not fully revealed since the state of art. After reading the manuscript, the impression was that the authors reported a large number of results in different areas, rather than focusing on creating a high-quality solution with a convincing comparative analysis of effectiveness. The manuscript appears blurry, and the results individually are not of significant scientific interest. A serious rework is required from the formulation of the problem and existing solutions review to the high-quality finalization in the discussion and conclusions section.
The Related work section should be added with CV methods based on:
Tsapin, D., Pitelinskiy, K., Suvorov, S. et al. Machine learning methods for the industrial robotic systems security. J Comput Virol Hack Tech (2023). https://doi.org/10.1007/s11416-023-00499-6
and industrial robotic manipulators based on:
Krakhmalev, O.; Krakhmalev, N.; Gataullin, S.; Makarenko, I.; Nikitin, P.; Serdechnyy, D.; Liang, K.; Korchagin, S. Mathematics Model for 6-DOF Joints Manipulation Robots. Mathematics 2021, 9, 2828. https://doi.org/10.3390/math9212828
In terms of CV methods, an additional section is required with a description of datasets and data preprocessing tools, as well as a significant expansion of the MS comparative performance analysis, taking into account the Related work section amendment. A link to datasets and software is highly desirable.
Author Response
Thank you very much for taking the time to review this manuscript. Please find the detailed responses below and the corresponding corrections highlighted changes in the re-submitted files.Please see the attachment.

Round 2
Reviewer 3 Report
Comments and Suggestions for Authors
This is appropriate